# Reintervention after Thoracic Endovascular Aortic Repair of Uncomplicated Type B Aortic Dissection

**DOI:** 10.3390/jcm12041418

**Published:** 2023-02-10

**Authors:** Li Cheng, Dongqiao Xiang, Shan Zhang, Chuansheng Zheng, Xiaoyan Wu

**Affiliations:** 1Department of Pediatrics, Union Hospital, Tongji Medical College, Huazhong University of Science and Technology, Wuhan 430022, China; 2Department of Radiology, Union Hospital, Tongji Medical College, Huazhong University of Science and Technology, Wuhan 430022, China; 3Hubei Province Key Laboratory of Molecular Imaging, Wuhan 430022, China; 4Department of Vascular Surgery, Union Hospital, Tongji Medical College, Huazhong University of Science and Technology, Wuhan 430022, China

**Keywords:** aortic dissection, endovascular repair, reintervention

## Abstract

Background: Data are scarce regarding the incidence, reasons, potential risk factors, and long-term outcomes of reintervention after thoracic endovascular aortic repair (TEVAR) in patients with uncomplicated type B aortic dissection (TBAD). Methods: Between January 2010 and December 2020, 238 patients with uncomplicated TBAD who received TEVAR were analyzed retrospectively. The clinical baseline data, aorta anatomy, dissection characteristics, and details of the TEVAR procedure were evaluated and compared. A competing-risk regression model was used to estimate the cumulative incidences of reintervention. The multivariate Cox model was used to identify the independent risk factors. Results: The mean follow-up time was 68.6 months. A total of 27 (11.3%) cases of reintervention were observed. The competing-risk analyses showed that the 1-, 3-, and 5-year cumulative incidences of reintervention were 5.07%, 7.08%, and 14.0%, respectively. Reasons for reintervention included endoleak (25.9%), aneurysmal dilation (22.2%), retrograde type A aortic dissection (18.5%), distal stent-graft-induced new entry and false lumen expansion (18.5%), and dissection progression and/or malperfusion (14.8%). Multivariable Cox analysis demonstrated that a larger initial maximal aortic diameter (Hazard ratio [HR], 1.75; 95% Confidence interval [CI], 1.13–2.69, *p* = 0.011) and increased proximal landing zone oversizing (HR, 1.07; 95% CI, 1.01–1.47, *p* = 0.033) were the significant risk factors for reintervention. Long-term survival rates were comparable between patients with or without reintervention (*p* = 0.915). Conclusions: Reintervention after TEVAR in patients with uncomplicated TBAD is not uncommon. A larger initial maximal aortic diameter and excessive proximal landing zone oversizing are associated with the second intervention. Reintervention does not significantly affect long-term survival.

## 1. Introduction

Aortic dissection is a catastrophic cardiovascular disease. Thoracic endovascular aortic repair (TEVAR) has altered the management algorithm of type B aortic dissection (TBAD) since its introduction by Dake et al. in 1994 [1]. With lower mortality rates and fewer complications, TEVAR has replaced traditional open surgery as the treatment of choice for patients with complicated TBAD [2]. For uncomplicated TBAD, the preferred treatment is currently under debate. Some guidelines recommend the best medical treatment (BMT) to control blood pressure and slow the heart rate as first-line therapy [3,4]. However, published evidence suggests that TEVAR results in significantly better long-term survival and aortic remodeling than BMT in patients with uncomplicated TBAD [5,6,7,8].

Regrettably, TEVAR is not a once-and-for-all solution. The long-term benefits of TEVAR may be attenuated or even offset by postoperative adverse events, such as endoleak, stroke, retrograde type A aortic dissection (RTAD), stent-graft migration, distal stent-graft-induced new entry (dSINE), and aneurysmal degeneration. These are the biggest obstacles to TEVAR becoming a first-line treatment for uncomplicated TBAD. In most situations, these complications need reintervention to save lives or to achieve better aortic remodeling.

In the literature published to date, some studies have addressed the issue of reintervention after TEVAR [9,10,11,12]. However, the majority of these are based on populations with complicated TBAD, and/or mixed cases of dissection and aneurysm, or involve a short follow-up period. The purpose of this study was to evaluate the incidence, reasons, potential risk factors, and long-term outcomes of reintervention in patients with uncomplicated TBAD.

## 2. Materials and Methods

### 2.1. Study Population

Between January 2010 and December 2020, 1128 patients were diagnosed with TBAD at the Union Hospital of Huazhong University of Science and Technology. The local ethical review board approved the study, and informed consent was waived because of its retrospective nature. Patients with TBAD confirmed by arterial computed tomography angiography (CTA) and who were treated with TEVAR were included. Uncomplicated TBAD was defined as dissection without rupture, malperfusion syndrome, refractory pain, or rapid aortic expansion at onset or during a patient’s presentation at the hospital [13]. Finally, 238 patients diagnosed with uncomplicated TBAD who received TEVAR were enrolled in the study. The flowchart of patient selection is shown in Figure 1.

### 2.2. TEVAR Procedure

Indications for TEVAR are usually considered to include an initial maximum aortic diameter ≥40 mm, an initial false lumen diameter ≥22 mm, a primary entry tear diameter ≥10 mm, and a patent or partially thrombosed false lumen [4,14,15]. The need for TEVAR and its timing were discussed among the surgical decision-making team in all cases.

Details of the TEVAR procedure were described in a previous study [16]. The procedure was typically performed in a hybrid operating room under local or general anesthesia by experienced endovascular and vascular surgeons, cardiologists, or interventional radiologists. The size of each thoracic stent graft was selected based on the diameter of the normal aorta proximal to the dissection. Proximal oversizing was usually chosen from a range of 1–10%, and the distal diameter was left to the discretion of the surgeon. We do not routinely administer antithrombotic therapy postoperatively.

### 2.3. Data Collection and Follow-Up

Baseline clinical data and details about TEVAR were collected retrospectively by searching the electronic medical record system. The anatomical features of the aorta were evaluated and measured in preoperative CTA images based on the centerline of flow in the Syngo.via post-processing workstation (Siemens Healthcare, Forchheim, Germany). Postoperative aortic-related events were evaluated using follow-up CTA images, which were taken routinely at 1-, 3-, 6-, and 12-month intervals after the primary TEVAR procedure, and yearly thereafter. Details of reintervention were obtained from patient readmission records. The follow-up period for the present study ended on 30 June 2022.

### 2.4. Endpoints and Definitions

The outcomes were the incidence, reasons, potential risk factors, and outcomes of reintervention after primary TEVAR during follow-up. Reintervention was defined as a second intervention for the management of complications after the primary TEVAR, not including the second-stage operation [17]. The coverage ratio of the thoracic aorta was defined as the ratio of the length of thoracic aortic stent coverage to the total length of the thoracic aorta. Aneurysmal degeneration was defined as an increase in the total aortic diameter of >5.5 cm or growth of ≥0.5 cm in a 6-month period [18].

### 2.5. Statistical Analysis

Continuous variables were expressed as the mean ± standard deviation or the median with the interquartile range, whereas categorical variables were presented as the frequency and percentages. Statistical comparisons were made with the χ2 or Fisher exact tests for categorical variables and Student’s *t* or Mann–Whitney U tests for continuous variables. The Cox proportional hazards model was used to identify risk factors for reintervention. *p* < 0.1 in the univariate analysis was selected for entry into the multivariable regression analysis. Reinterventions were evaluated by a competing-risk analysis with mortality as the competing event, and between-group differences were assessed using the Fine–Gray test for estimates of cumulative incidences. All statistical tests were two-sided, and the significance level was <0.05. Statistical analyses were performed using R 4.1.2 (R Development Core Team, Vienna, Austria).

## 3. Results

### 3.1. Baseline Characteristics

In total, 238 patients with uncomplicated TBAD treated with TEVAR were included in this study; among them, 27 (11.3%) patients received reintervention during the follow-up. The baseline characteristics are listed in Table 1. The mean age was 52.6 ± 10.9 years in the non-intervention group and 54.4 ± 9.1 years in the intervention group (*p* = 0.397). Hypertensive patients accounted for 65.4% of the total in the non-reintervention group and 66.7% in the reintervention group (*p* = 0.896).

### 3.2. Anatomical Features and TEVAR Procedure

Anatomical features of the thoracic aorta and details about the TEVAR procedure are listed in Table 2. The median initial maximal aortic diameter was 3.6 cm in the non-reintervention group and 3.5 cm in the reintervention group (*p* = 0.142). The mean oversizing at the proximal landing zone was 4.57 ± 4.26% in the non-reintervention group and 6.68 ± 6.20% in the reintervention group (*p* = 0.067). There were no significant differences in the stent graft length, coverage ratio of thoracic aorta, or proximal landing zone between the two groups.

Technical success was achieved in all patients by the primary TEVAR procedure. A total of 269 stent grafts were implanted in the thoracic aortas of 238 patients. Among them, 209 patients (87.8%) received one stent graft, 27 (11.3%) received two stent grafts, and 2 (0.8%) received three stent grafts. Valiant (Medtronic, Inc, Minneapolis, MN, USA) was the most used brand at 46.1% (*n* = 124), followed by Ankura (Lifetech Scientific, Shenzhen, China) at 24.5% (*n* = 66).

### 3.3. In-Hospital and 30-Day Outcomes 

In-hospital and 30-day outcomes are reported in Table 3. One patient in the non-reintervention group died because of a sudden stroke and RTAD 22 days after primary TEVAR. The difference in the 30-day mortality rate between the two groups was not statistically significant (*p* = 1.000). The total numbers of in-hospital and 30-day adverse events were comparable between the two groups (7.6% vs. 18.5%, *p* = 0.072). However, the incidence of RTAD was significantly higher in the reintervention group than in the non-reintervention group (7.4% vs. 0.5%, *p* = 0.035). Two patients in the reintervention group developed RTAD, both of whom underwent total arch replacement. Type I endoleaks were the most common adverse events, with a total of eleven type I endoleaks observed during the first 30 days, nine of which were mild and did not require reintervention.

### 3.4. Incidence and Reasons

The mean follow-up time was 63.7 ± 25.9 months in the non-reintervention group and 73.4 ± 25.8 months in the reintervention group (*p* = 0.069). A total of 27 (11.3%) cases of reintervention were observed during the follow-up period. Considering death as a competing risk, cumulative incidence estimates suggest that the 1-, 3-, and 5-year cumulative incidences of reintervention were 5.07%, 7.08%, and 14.0%, respectively (Figure 2). The reasons for reinterventions, as well as their type and timing, are listed in Table 4. Reasons for reintervention included endoleak (25.9%), RTAD (18.5%), dSINE and false lumen expansion (18.5%), aneurysmal dilation (non-dSINE causes) (22.2%), and dissection progression and/or malperfusion (14.8%).

### 3.5. Risk Factors and Outcome of Reintervention

The results of the Cox hazard regression analyses are summarized in Table 5. Univariate analysis and multivariable Cox hazard regression analysis demonstrated that a larger maximal aortic diameter (hazard ratio [HR], 1.75; 95% confidence interval [CI], 1.13–2.69, *p* = 0.011) and increased proximal landing zone oversizing (HR, 1.07; 95% CI, 1.01–1.47, *p* = 0.033) were the significant risk factors for reintervention. Kaplan–Meier survival analysis showed that 1-, 3-, 5-, and 10-year overall survival rates in the intervention group were 100%, 100%, 95.5%, and 75.5%, respectively, with corresponding figures of 99.5%, 97.6%, 95.7%, and 72.0% for the non-intervention group (Figure 3). A log-rank test showed that the presence or absence of reintervention had no significant effect on overall survival (*p* = 0.915).

## 4. Discussion

The present study is one of the first to use a large cohort to report the incidence, reasons, risk factors, and outcomes of reintervention after TEVAR in a subset of uncomplicated TBAD cases with an average follow-up of more than 5 years. To objectively assess the incidence, a competing-risk model with all-cause mortality as a competing event was used to determine the cumulative incidence of reintervention. To explore potential risk factors, we considered multiple variables including clinical baseline data, aorta anatomy, dissection characteristics, and details of the TEVAR procedure.

In a previously published meta-analysis of 27 studies involving 2403 patients with aortic dissection, the pooled overall incidence of reintervention was 15.0% during 33.7 months of follow-up [17]. In the present study, 11.3% of patients required reintervention and the 3- and 5-year cumulative incidences of reintervention were 7.08% and 14.0%. These results were lower than the results reported in the above systematic review. A possible explanation is that this study only focused on uncomplicated TBAD, whereas the data reported in the literature mostly concern mixed cases of all dissections and/or aneurysms. Uncomplicated TBAD may be less severe in pathological processes than complicated TBAD and aneurysms.

In this study, the reasons for reintervention in descending order were as follows: Endoleak, aneurysmal dilation (non-dSINE causes), RTAD, dSINE and false lumen expansion, and dissection progression and/or malperfusion. This finding is consistent with the results of a previous study [17]. Interestingly, the incidence of these indications did not differ much but was close. Endoleak, especially type I endoleak, is still the most common reason for reintervention after TEVAR. Nozdrzykowski et al. [19] also reported type I endoleak as the most frequent indication for a secondary procedure (26.8%; *n* = 15/56). The use of scallop, fenestration, and chimney techniques, as well as steep aortic arches, too-short landing zones, and oversizing (too large or too small), have all been reported as causative factors for type I endoleak [10,11,20]. In the present study, of the six cases of endoleak that underwent reintervention, two cases progressed after 30 days of detection, and four cases were newly developed during follow-up. Timely intervention (e.g., cuff, candy-plug, coil and/or plug embolization, and STABILISE techniques) is necessary when persistent moderate or severe endoleaks are found, and all endoleaks in this study achieved good long-term outcomes after reintervention. Aneurysmal dilation is a chronic process that may be difficult to detail as intrinsic factors of the disease may be responsible. The length of the covered segment seems to play a major role. Some studies have reported that covering the entire thoracic aorta with stents can prevent the occurrence of aneurysmal dilation and promote aortic remodeling [21,22].

RTAD and dSINE share the same pathogenic mechanism, but RTAD is more dangerous and often requires open surgical repair. Although there are many possible reasons for RTAD after TEVAR, including excessive oversizing, proximal bare stents, acute dissection or connective tissue disorders, and inappropriate landing zones [11,23,24,25], excessive oversizing remains the most recognized risk factor in many studies. In the present study, multivariable regression analysis demonstrated that excessive proximal landing zone oversizing is a significant risk factor for reintervention, and this is consistent with the findings of previous studies [10,26]. Excessive oversizing can not only lead to RTAD but may also cause endoleak and dSINE [10,27]. Although the potential relationship between oversizing and reintervention has not been elucidated, the selection of oversizing is important. At present, the optimal oversizing for TBAD remains controversial. In our experience, an oversizing of approximately 5% is appropriate for TEVAR treatment of uncomplicated TBAD. In an integrated data study, Canaud et al. indicated that when oversizing was greater than 9%, the relative risk of RTAD increased by 14% for each percentage-point increase [25]. No in-stent thrombosis or stent folding was observed in this study. This also confirmed that it is difficult to form a thrombus in the thoracic aortic stent after TEVAR, and oversizing within 10% can effectively prevent the occurrence of stent folding [28].

The initial maximal aortic diameter is considered to be a predictor of poor prognosis in patients with uncomplicated TBAD treated with medical therapy alone. Early TEVAR intervention can thus be recommended for these high-risk patients. Interestingly, the present study showed that, even with TEVAR treatment, a larger initial maximal aortic diameter remained a significant risk factor for reintervention. Giles et al. also reported that a larger maximal aortic diameter at presentation was associated with secondary aortic intervention after TEVAR for aortic dissection [26]. Therefore, whether patients with larger initial maximal aortic diameters should decline open surgery or receive TEVAR with lifelong close surveillance is an issue that needs to be explored in future studies.

This study has some limitations. First, it is a single-center retrospective study, and potential selection bias is difficult to avoid in such research. Second, there are no standard guidelines for TEVAR and subsequent secondary interventions in uncomplicated TBAD, so our empirical decisions on TEVAR and reinterventions may have potentially influenced outcomes of uncomplicated TBAD. Third, although we tried to incorporate many features to find potential risk factors for reintervention, the findings reported here may yet represent only the tip of the iceberg.

## 5. Conclusions

The incidence of reintervention after TEVAR in patients with uncomplicated TBAD is not low. Larger initial maximal aortic diameter and excessive proximal landing zone oversizing were associated with reintervention after TEVAR. Multiple pathological factors contributed to reintervention. However, reintervention did not significantly affect long-term survival.

## Figures and Tables

**Figure 1 jcm-12-01418-f001:**
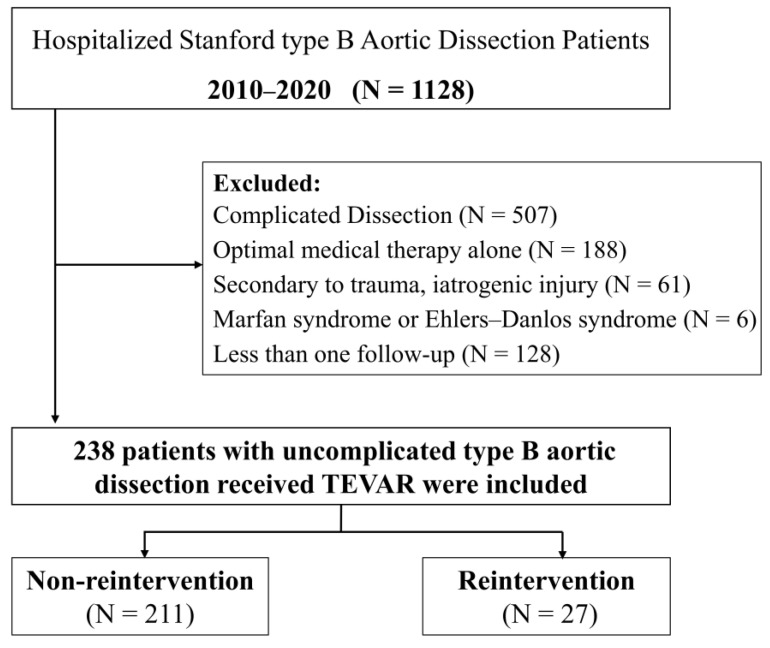
Flowchart of patient selection.

**Figure 2 jcm-12-01418-f002:**
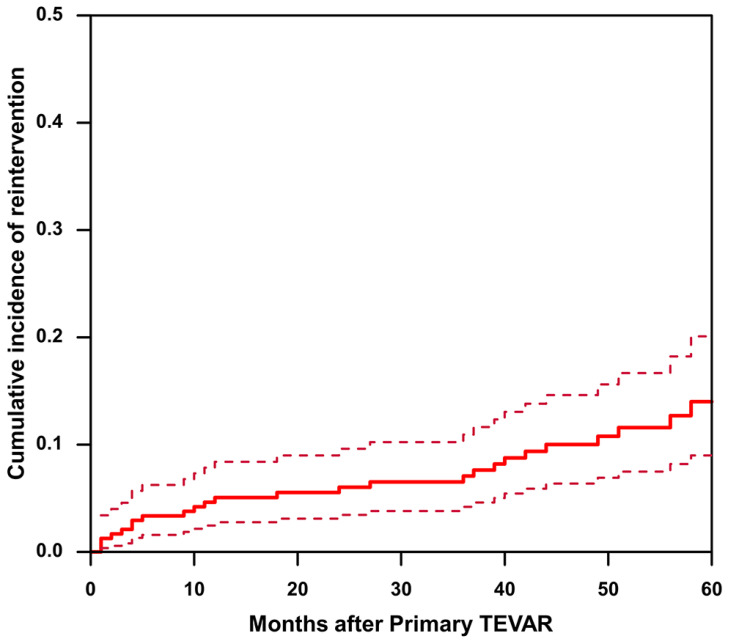
The cumulative incidence of reintervention after TEVAR in patients with uncomplicated type B aortic dissection. The solid line represents cumulative incidence curve. The dashed line represents the 95% confidence interval of the cumulative incidence.

**Figure 3 jcm-12-01418-f003:**
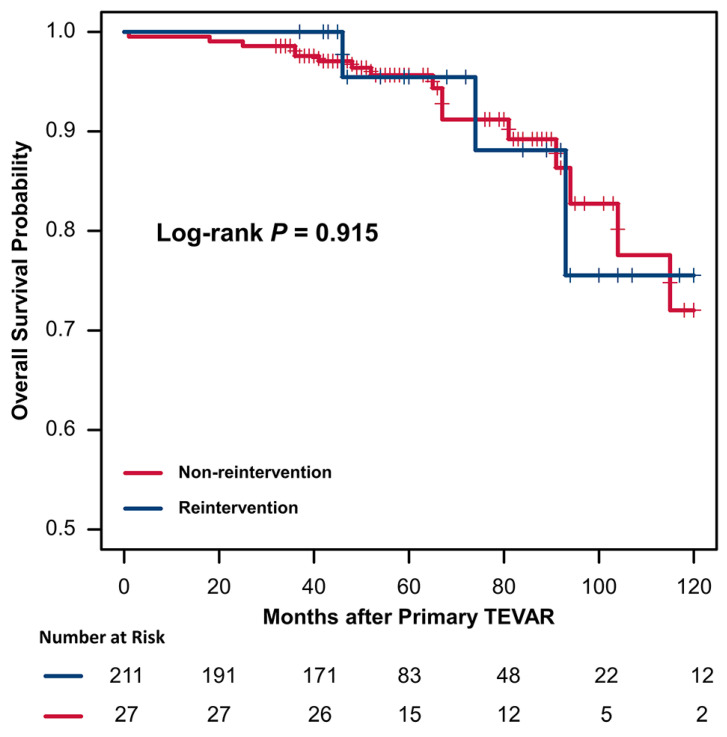
Kaplan–Meier curves of overall survival in patients with or without reintervention after TEVAR.

**Table 1 jcm-12-01418-t001:** Baseline covariate distribution.

Variable	Overall (*n* = 238)	Non-Reintervention (*n* = 211)	Reintervention (*n* = 27)	*p* Value
Age (y)	52.8 ± 10.7	52.6 ± 10.9	54.4 ± 9.1	0.397
Male	199 (83.6)	176 (83.4)	23 (85.2)	0.815
Hypertension	156 (65.5)	138 (65.4)	18 (66.7)	0.896
Smoking	111 (46.6)	97 (46.0)	14 (51.9)	0.564
Drinking	74 (31.1)	62 (29.4)	12 (44.4)	0.111
Diabetes mellitus	12 (5.0)	11 (5.2)	1 (3.7)	0.736
COPD	27 (11.3)	24 (11.4)	3 (11.1)	1.000
Renal insufficiency	28 (11.8)	27 (12.8)	1 (3.7)	0.218
Coronary artery disease	18 (7.6)	15 (7.1)	3 (11.1)	0.439
Hyperlipidemia	6 (2.5)	6 (2.8)	0 (0.0)	1.000
History of stroke	13 (5.5)	13 (6.2)	0 (0.0)	0.372
Atherosclerosis	83 (34.9)	70 (33.2)	13 (48.1)	0.124
Pleural effusion	45 (18.9)	38 (18.0)	7 (25.9)	0.323
TBAD duration				0.429
Acute aortic dissection	144 (60.5)	130 (61.6)	14 (51.9)	
Subacute aortic dissection	78 (32.8)	66 (31.3)	12 (44.4)	
Chronic aortic dissection	16 (6.7)	15 (7.1)	1 (3.7)	
Dissection morphology				0.263
Confined in thoracic aorta	36 (15.1)	30 (14.2)	6 (22.2)	
Extended to abdominal aorta	202 (84.9)	181 (85.8)	21 (77.8)	
False lumen patency				0.778
Patent false lumen	135 (56.7)	119 (56.4)	16 (59.3)	
Partial thrombosis	103 (43.3)	92 (43.6)	11 (40.7)	
Complete thrombosis	0 (0.0)	0 (0.0)	0 (0.0)	
SBP on admission (mmHg)	142 (127, 159)	142 (126, 159)	140 (127, 160)	0.950
DBP on admission (mmHg)	83 (75, 93)	83 (74, 91)	84 (78, 97)	0.335
Creatinine (μmol/L)	71.1 (62.5, 80.3)	71.0 (61.4, 81.1)	71.8 (65.1, 77.3)	0.878
ALT (U/L)	23.0 (16.0, 35.0)	24.0 (16.0, 35.0)	20.0 (16.0, 44.0)	0.833
AST (U/L)	20.0 (14.8, 27.0)	20.0 (14.0, 27.0)	20.0 (15.0, 27.0)	0.763
Symptoms at presentation				0.488
Chest and back pain	204 (85.7)	182 (86.3)	22 (81.5)	
Abdominal pain	22 (9.2)	18 (8.5)	4 (14.8)	
Other symptoms	12 (5.0)	11 (5.2)	1 (3.7)	

Values are mean ± standard deviation (SD) or *n* (%). TBAD—type B aortic dissection; COPD—chronic obstructive pulmonary disease; SBP—systolic blood pressure; DBP—diastolic blood pressure; ALT—alanine aminotransferase; AST—aspartate aminotransferase.

**Table 2 jcm-12-01418-t002:** Anatomical features of the thoracic aorta and details of the TEVAR procedure.

Variable	Non-Reintervention (*n* = 211)	Reintervention (*n* = 27)	*p* Value
Aortic arch classification			0.242
Type I	106 (50.2)	9 (33.3)	
Type II	63 (29.9)	10 (37.0)	
Type III	42 (19.9)	8 (29.6)	
Maximal aortic diameter (cm)	3.5 (3.2, 3.8)	3.6 (3.3, 4.5)	0.142
Proximal landing zone oversizing (%)	4.57 ± 4.26	6.68 ± 6.20	0.067
Proximal landing zone diameter (mm)	29.81 ± 2.68	29.94 ± 2.93	0.803
Main stent graft length (mm)	175.37 ± 24.43	177.04 ± 22.50	0.737
Length of covered thoracic aorta (mm)	177.68 ± 25.69	182.59 ± 26.11	0.351
Coverage ratio of thoracic aorta (%)	62.11 ± 9.07	63.22 ± 9.98	0.555
Stent graft proximal diameter (mm)	30.88 ± 2.73	31.44 ± 3.19	0.324
Proximal bare stent	205 (97.2)	25 (92.6)	0.226
Proximal landing zone			0.509
Zone 1	5 (2.4)	1 (3.7)	
Zone 2	147 (69.7)	17 (63.0)	
Zone 3	59 (28.0)	9 (33.3)	
Distance between LSA and PET (mm)	20.72 ± 13.27	23.31 ± 13.61	0.342
Intentional coverage of the LSA	48 (22.7)	8 (29.6)	0.427
Chimney graft of the LSA	8 (3.8)	1 (3.7)	0.982
Chimney graft of the LCCA	5 (2.4)	1 (3.7)	0.518
Thoracic stent graft brand	235	34	
Valiant (Medtronic, Inc, Minneapolis, MN, USA)	109 (46.4)	15 (44.2)	0.804
E-vita (JOTEC GmbH, Hechingen, Germany)	24 (10.2)	2 (5.9)	0.752
Zenith TX2(Cook, Bloomington, IN, USA)	7 (3.0)	1 (2.9)	1.000
Hercules (Microport, Shanghai, China)	37 (15.7)	8 (23.5)	0.256
Ankura (Lifetech Scientific, Shenzhen, China)	58 (24.7)	8 (23.5)	0.884

Values are mean ± standard deviation (SD) or *n* (%). TEVAR—thoracic endovascular aortic repair; PET—primary entry tear; LSA—left subclavian artery; LCCA—left common carotid artery.

**Table 3 jcm-12-01418-t003:** In-hospital and 30-day outcomes.

Variable	Non-Reintervention (*n* = 211)	Reintervention (*n* = 27)	*p* Value
Hospital stay (days)	10.1 ± 3.3	10.4 ± 3.3	0.602
30-day mortality	1 (0.5)	0 (0.0)	1.000
Adverse events	16 (7.6)	5 (18.5)	0.072
Acute renal failure	2 (0.9)	0 (0.0)	1.000
Type I endoleak	9 (4.3)	2 (7.4)	0.361
Retrograde type A aortic dissection	1 (0.5)	2 (7.4)	0.035
Stroke	1 (0.5)	0 (0.0)	1.000
Organ failure	3 (1.4)	1 (3.7)	0.384

Values are mean ± standard deviation (SD) or *n* (%).

**Table 4 jcm-12-01418-t004:** Indications and descriptions of reinterventions.

Indications	Number (%)	Type of Reintervention	Time to Reintervention (Months after Initial TEVAR)
Endoleaks	7 (25.9%)	7 endovascular	10.0 (3.0, 42.0)
Type I endoleak	6 (22.2%)	6 endovascular	20.0 (2.5, 46.0)
Type III endoleak	1 (3.7%)	1 endovascular	10.0
RTAD	5 (18.5%)	5 open surgery	9.0 (1.0, 68.0)
dSINE and false lumen expansion	5 (18.5%)	5 endovascular	5.0 (3.0, 20.0)
Aneurysmal dilation (non-dSINE causes)	6 (22.2%)	5 endovascular; 1 open surgery	47.5 (39.3, 64.5)
Dissection progression and/or malperfusion	4 (14.8%)	3 endovascular; 1 open surgery	26.5 (11.8, 46.5)

Time to reintervention expressed as median [25th, 75th percentile]. RTAD—retrograde type A aortic dissection; dSINE—distal stent-graft-induced new entry.

**Table 5 jcm-12-01418-t005:** Univariate and multivariate Cox hazard analysis of reintervention.

Variable	Univariate Analysis		Multivariate Analysis	
	HR (95% CI)	*p* Value	HR (95% CI)	*p* Value
Age (y)	1.02 (0.98–1.06)	0.347		
Male	1.05 (0.36–3.04)	0.932		
Hypertension	0.97 (0.44–2.17)	0.946		
TBAD duration				
Acute aortic dissection	Ref			
Subacute aortic dissection	1.62 (0.75–3.49)	0.222		
Chronic aortic dissection	0.52 (0.07–3.96)	0.528		
Dissection morphology	1.80 (0.72–4.49)	0.209		
Patent false lumen	1.16 (0.54–2.51)	0.700		
Aortic arch classification				
Type I	Ref			
Type II	1.65 (0.69–4.07)	0.278		
Type III	2.13 (0.81–5.52)	0.121		
Maximal aortic diameter (mm)	1.65 (1.08–2.52)	0.022	1.75 (1.13–2.69)	0.011
Proximal landing zone diameter (mm)	1.01 (0.87–1.16)	0.922		
Proximal landing zone oversizing (%)	1.06 (0.99–1.13)	0.063	1.07 (1.01–1.47)	0.033
Main stent graft length (mm)	1.01 (0.99–1.02)	0.534		
Length of covered thoracic aorta (mm)	1.10 (0.96–1.27)	0.169		
Coverage ratio of thoracic aorta (%)	1.02 (0.98–1.06)	0.363		
Stent graft proximal diameter (mm)	1.05 (0.91–1.20)	0.516		
Proximal bare stent	1.83 (0.42–8.04)	0.425		
Proximal landing zone				
Zone 1	Ref			
Zone 2	0.85 (0.11–6.56)	0.878		
Zone 3	0.99 (0.12–8.01)	0.996		
Distance between LSA and PET (mm)	1.02 (0.99–1.04)	0.250		
Intentional coverage of the LSA	0.83 (0.36–1.89)	0.652		
Chimney graft of the LSA	0.77 (0.10–5.69)	0.795		
Chimney graft of the LCCA	0.56 (0.08–4.15)	0.570		

HR—Hazard ratio; CI—confidence interval. TBAD—type B aortic dissection; PET—primary entry tear; LSA—left subclavian artery; LCCA—left common carotid artery.

## Data Availability

The data and/or related materials of this study are available from the corresponding author upon reasonable request.

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
