# Peer review of "Reintervention after Thoracic Endovascular Aortic Repair of Uncomplicated Type B Aortic Dissection"

_jcm, 2023, doi:10.3390/jcm12041418_

Round 1

Reviewer 1 Report

Is a good paper with a large casistic.Some abbreviations in the text must be explain.In the series no cases of partial or total thrombosis of the endograft are reported and no cases of endograft infolding.Can you explain why these complications also described in the literature were not observed?

In any cases the issues relating to these complications should be addressed in the discussion.As reported in the text the oversizing is one of the maior problem related to the complications, principally  the infolding and the thrombosis. Wath you think about the use of conic endograft to better obtain a remodelling of the aorta in the distal portion and to avoid oversizing in this site?

In the conclusion also the oversizing should be cited as cause of possible complications.

Author Response

Dear Editors and Reviewers:

Thank you very much for offering the opportunity to revise our manuscript “Reintervention after thoracic endovascular aortic repair of uncomplicated type B aortic dissection”. We really appreciated and cherished this opportunity and read editors’ and reviewers’ comments carefully. We tried our best to revise our manuscript according to these comments. You will find below the point-by-point response.

# Reviewer 1

(x) Moderate English changes required

Response: Thank you very much for taking the time to review our manuscript and for your valuable comments. As you suggested, we have addressed this issue by using the English editing services officially provided by MDPI. You will find that the text and expression of the article have been heavily modified to make it more accurate and concise. We have carefully checked our manuscript and made many revisions, but there are unavoidable omissions. Looking forward to your further guidance. Thank you again for helping us improve our article.

Are all the cited references relevant to the research? ( )  (x) ( )  ( )

Response: Thank you for your guidance. We attach great importance to your evaluation. We reviewed all literature to ensure its relevance to this study. In addition, we have appropriately added relevant references.

Are the results clearly presented? ( )  (x) ( )  ( )

Response: Thank you for pointing out this issue. On the one hand, we have modified the content of the Results section to make it more clear and readable; on the other hand, we have carried out language polishing so that the results can be clearly expressed.

Are the conclusions supported by the results? ( ) (x) ( )  ( )

Response: Thank you for your guidance. We have made appropriate revisions to the conclusions and look forward to your further guidance.

Comments and Suggestions for Authors

Is a good paper with a large casistic. Some abbreviations in the text must be explain. In the series no cases of partial or total thrombosis of the endograft are reported and no cases of endograft infolding. Can you explain why these complications also described in the literature were not observed?

Response: Thank you very much for your patient instructions and suggestions. As you suggested, we have explained the meaning of all abbreviations in the text. In our cohort, partial or total thrombosis within the endograft and endograft infolding were not observed. In-stent thrombosis is a common complication after endovascular repair of branch vessels, but it is rare in endovascular treatment of thoracic aortic dissection. Because of the large aortic blood flow, fast velocity, high pressure, and wide aortic stent diameter, it is difficult to form thrombus in a thoracic stent. In contrast, branch stent thrombosis is often observed in studies of endovascular treatment of diseases such as abdominal aortic aneurysm and lower extremity arterial embolism [1-3]. This study focused on endovascular repair of thoracic aorta with aortic dissection, and no partial or complete thrombosis was observed within the stent. Endograft infolding folding has been extensively reported in the early years for either branch stents or thoracic aortic stents. Many studies have shown that stent folding is associated with the use of too large a stent oversizing [4-5]. Based on nearly 30 years of clinical experience, the oversizing we usually choose is 5%-10% for type B aortic dissection, so it is not easy to cause the endograft to fold.

[1] Troisi N, Torsello G, Donas KP, et al. Endurant stent-graft: a 2-year, single-center experience with a new commercially available device for the treatment of abdominal aortic aneurysms. J Endovasc Ther. 2010;17(3):439-448.

[2] Liu MY, Jiao Y, Liu J, et al. Hemodynamic Parameters Predict In-stent Thrombosis After Multibranched Endovascular Repair of Complex Abdominal Aortic Aneurysms: A Retrospective Study of Branched Stent-Graft Thrombosis. Front Cardiovasc Med. 2021;8: 654412.

[3] Laird JR Jr, Yeo KK, Rocha-Singh K, et al. Excimer laser with adjunctive balloon angioplasty and heparin-coated self-expanding stent grafts for the treatment of femoropopliteal artery in-stent restenosis: twelve-month results from the SALVAGE study. Catheter Cardiovasc Interv. 2012;80(5):852-859.

[4] Kasirajan K, Dake MD, Lumsden A, et al. Incidence and outcomes after infolding or collapse of thoracic stent grafts. J Vasc Surg. 2012;55(3):652-658.

[5] Mestres G, Uribe JP, García-Madrid C, et al. The best conditions for parallel stenting during EVAR: an in vitro study. Eur J Vasc Endovasc Surg. 2012;44(5):468-473.

In any cases the issues relating to these complications should be addressed in the discussion. As reported in the text the oversizing is one of the maior problem related to the complications, principally the infolding and the thrombosis. Wath you think about the use of conic endograft to better obtain a remodelling of the aorta in the distal portion and to avoid oversizing in this site?

Response: Thank you for your guidance. We discussed stent folding and thrombus issues in the Discussion in revision. Thank you for asking this thought-provoking question. Our previous study specifically explored the use of tapered stents (conic endograft) to prevent distal stent graft-induced new entry caused by excessive distal oversizing [6]. In our center, for patients with relatively large thoracic aortic taper, tapered stents are used to prevent excessive distal oversizing. Our previous study showed that the use of a tapered stent graft with a >4-mm taper could help prevent distal stent graft-induced new entry in patients with a high taper ratio.

Changes: “No in-stent thrombosis or stent folding was observed in this study. This also confirmed that it is difficult to form thrombus in the thoracic aortic stent after TEVAR, and oversizing within 10% can effectively prevent the occurrence of stent folding.Revised manuscript Page 10 lines 242-245

[6] Xiang D, Chai B, Gui Y, et al. Risk factors for distal stent graft-induced new entry after endovascular repair in uncomplicated type B aortic dissection. J Vasc Surg. 2023;77(1):37-45.e1.

In the conclusion also the oversizing should be cited as cause of possible complications.

Response: Thank you for pointing out this issue. Following your suggestion, we put the oversizing as cause of possible complications in the conclusion. Thanks again for taking the time to help us improve our articles. Your suggestions and questions have important guiding significance for us to modify the manuscript. Looking forward to your further guidance.

Changes: “Larger initial maximal aortic diameter and excessive proximal landing zone oversizing were associated with reintervention after TEVAR.Revised manuscript Page 10 lines 266

Reviewer 2 Report

Dear Authors,

First of all i would like to congratulate you for your study. TYpe B aortic dissection  is a very  interesting  topic of discussion between surgeons. However  you have to clarify some parts of the text and to change some phrases in order to improve your paper.

Pg 1 39-> Replace fewer with less

Pg 1 49-> Change the phrase in plural... are the biggest obstacles to become

Table 1. 14 patients underwent TEVAR in acute period. You have to explain the reasons why you operated them in acute and not in subacute period.We all know that adverse events are more common in acute  period. Did your patients were under proper anti-impulse and anti-hypertensive treatment during hospitalization?

Table 2. Please explain why did you use stent grafts with proximal bare stent which are associated with higher incidence of retrograde dissection and not covered proximal endografts?

Please refer in your manuscript what type of antithrombotic treatment did you give to your patients postoperatively.

pg 6. You noticed 11 EL1 post-operatively, 9 of them where mild. What are your criteria to define and EL1 mild? As we know EL1 is a phenomenon that needs immediate repair and we can't be in a " wait to see situation". You have to report the 6 IL1 that you treated were new during follow up or were they previously detected?

Discussion. Please i would like a comment in the discussion section regarding the Weissler study reporting that  initial TEVAr was not associated with improved survival,reduced-aortic related interventions-hospitalizationd during  1-2-5 years follow up. This will incerease citation of your paper.

Author Response

# Reviewer 2

Dear Editors and Reviewers:

Thank you very much for offering the opportunity to revise our manuscript “Reintervention after thoracic endovascular aortic repair of uncomplicated type B aortic dissection”. We really appreciated and cherished this opportunity and read editors’ and reviewers’ comments carefully. We tried our best to revise our manuscript according to these comments. You will find below the point-by-point response.

 Open Review

(x) English language and style are fine/minor spell check required
Response: Thank you very much for taking the time to review our manuscript and for your valuable comments. As you suggested, we have addressed this issue by using the English editing services officially provided by MDPI. You will find that the text and expression of the article have been heavily modified to make it more accurate and concise. We have carefully checked our manuscript and made many modifications, but there are unavoidable omissions. Looking forward to your further guidance. Thank you again for helping us improve our article.

Comments and Suggestions for Authors

Dear Authors,

First of all i would like to congratulate you for your study. Type B aortic dissection is a very interesting topic of discussion between surgeons. However you have to clarify some parts of the text and to change some phrases in order to improve your paper.

Pg 1 39-> Replace fewer with less

Response: Thank you for your guidance. As you suggested, we have replaced fewer with less.

Pg 1 49-> Change the phrase in plural... are the biggest obstacles to become

Response: Thank you for your careful guidance. We have revised it in the manuscript.

Table 1. 14 patients underwent TEVAR in acute period. You have to explain the reasons why you operated them in acute and not in subacute period. We all know that adverse events are more common in acute period. Did your patients were under proper anti-impulse and anti-hypertensive treatment during hospitalization?

Response: Thank you for asking this thought-provoking question. As you said, many studies [1-2] have shown that adverse events are more common in acute period. In our previous study [3], we specially investigated the impact of acute (1-14 days) and subacute (15-90 days) TEVAR on the long-term outcomes of uncomplicated TBAD, and the results also showed that there were fewer postoperative complications in the subacute period. We agree with your point about performing TEVAR in the subacute phase for more favorable results. However, regarding the specific intervention time, we think it is usually 2 weeks after presentation. Because the concept of subacute intervention was not widely recognized during the study period (2010-2020), there was no intentional delay to subacute intervention during the study period. However, in recent years, we have made more recommendations for patients to intervene in the subacute phase to reduce the risk of related complications. Best medical therapy with proper anti-impulse and anti-hypertensive during hospitalization is the basic treatment for uncomplicated TBAD. As described in our previous study [4], optimal drug therapy is used in all patients diagnosed with aortic dissection, regardless of subsequent treatment modalities. We look forward to communicating with you further on this issue.

[1] Torrent DJ, McFarland GE, Wang G, et al. Timing of thoracic endovascular aortic repair for uncomplicated acute type B aortic dissection and the association with complications. J Vasc Surg. 2021;73(3):826-835.

[2] Desai ND, Gottret JP, Szeto WY, et al. Impact of timing on major complications after thoracic endovascular aortic repair for acute type B aortic dissection. J Thorac Cardiovasc Surg. 2015;149(2 Suppl): S151-S156.

[3] Xiang D, Wu F, Chen L, et al. Timing of endovascular repair impacts long-term outcomes of uncomplicated acute type B aortic dissection. J Vasc Surg. 2022;75(3):851-860.e3.

[4] Xiang D, Kan X, Liang H, et al. Comparison of mid-term outcomes of endovascular repair and medical management in patients with acute uncomplicated type B aortic dissection. J Thorac Cardiovasc Surg. 2021;162(1):26-36.e1.

Table 2. Please explain why did you use stent grafts with proximal bare stent which are associated with higher incidence of retrograde dissection and not covered proximal endografts?

Response: Thank you for asking these valuable questions. As you said, many studies in recent years have shown that the bare stent at the proximal end of the stent is related to the occurrence of retrograde type A dissection. However, during the study period, the vast majority of thoracic aortic stents were tipped with bare metal for increased anchoring strength, such as Valiant (Medtronic, Inc, Minneapolis, MN), E-vita (JOTEC GmbH, Hechingen, Germany), Hercules (Microport, Shanghai, China) and Ankura (Lifetech Scientific, Shenzhen, China). Only a few types, such as Zenith TX2 (Cook Medical, Bloomington, IN), do not have bare metal stent at the head end. So, we have to use stent grafts with proximal bare stent. On the other hand, the association between proximal bare stents and retrograde type A dissections remains to be further studied. As concluded by Canaud’s systematic review [5], the incidence of RTAD was not significantly different for endografts with proximal bare stent (2.8%) or nonbare stent (1.9%) (P=0.1298). Looking forward to further communication with you on this issue.

[5] Canaud L, Ozdemir BA, Patterson BO, Holt PJ, Loftus IM, Thompson MM. Retrograde aortic dissection after thoracic endovascular aortic repair. Ann Surg. 2014;260(2):389-395.

Please refer in your manuscript what type of antithrombotic treatment did you give to your patients postoperatively.

Response: Thank you for your guidance. We do not routinely administer antithrombotic therapy postoperatively. In-stent thrombosis is a common complication after endovascular repair of branch vessels, but it is rare in endovascular treatment of thoracic aortic dissection. Because of the large aortic blood flow, fast velocity, high pressure, and wide aortic stent diameter, it is difficult to form thrombus in the thoracic stent. In contrast, antithrombotic therapy would slow or even hinder the process of thrombus of false lumen. It has almost become a consensus that false lumen thrombosis is a protective factor preventing aortic growth and aortic-related death. A study by Faure et al. [6] showed that anticoagulant therapy was a significant factor for reintervention after TEVAR in patients with complicated type B aortic dissection.

Changes: “We do not routinely administer antithrombotic therapy postoperatively.Revised manuscript Page 3 line 84

[6] Faure EM, Canaud L, Agostini C, et al. Reintervention after thoracic endovascular aortic repair of complicated aortic dissection. J Vasc Surg. 2014;59(2):327-333.

pg 6. You noticed 11 EL1 post-operatively, 9 of them where mild. What are your criteria to define and EL1 mild? As we know EL1 is a phenomenon that needs immediate repair and we can't be in a " wait to see situation". You have to report the 6 IL1 that you treated were new during follow up or were they previously detected?

Response: Thank you for your thoughtful suggestions. What we call mild type I endoleak is a small amount of contrast medium flowing into the false lumen observed on CTA images. Although there is currently no clear definition of mild endoleaks, previous studies [7] have shown that endoleaks with a maximum diameter of less than 15 mm can be safely observed first. Because some mild type I endoleaks can be absorbed and disappeared by themselves. Therefore, based on our experience, the strategy of observation and close follow-up can be adopted for small amounts of endoleak after TEVAR for aortic dissection. If the increase of endoleak is found during follow-up, prompt treatment is required. The main endovascular therapeutic options for EL1 include EndoAnchors, aortic cuffs and embolization [8]. In the present study, of the six cases of endoleak that underwent reintervention, two cases progressed after 30 days of detection, and four cases were newly developed during follow-up.

Changes: “In the present study, of the six cases of endoleak that underwent reintervention, two cases progressed after 30 days of detection, and four cases were newly developed during follow-up. Revised manuscript Page 10 lines 220-222

[7] Timaran CH, Ohki T, Rhee SJ, et al. Predicting aneurysm enlargement in patients with persistent type II endoleaks. J Vasc Surg. 2004;39(6):1157-1162.

[8] Ameli-Renani S, Pavlidis V, Morgan RA. Secondary Endoleak Management Following TEVAR and EVAR. Cardiovasc Intervent Radiol. 2020;43(12):1839-1854.

Discussion. Please i would like a comment in the discussion section regarding the Weissler study reporting that initial TEVAr was not associated with improved survival, reduced-aortic related interventions-hospitalizationd during 1-2-5 years follow up. This will incerease citation of your paper.

Response: Thank you for providing us with this valuable literature [8]. This comment is important to strengthen the evidence for TEVAR treatment of uncomplicated TBAD. We think it is more appropriate to place it in the introduction. We cited it in the Introduction (See References 8). Thanks again for taking the time to help us improve our articles. Your suggestions and questions have important guiding significance for us to modify the manuscript. Looking forward to your further guidance.

[8] Weissler EH, Osazuwa-Peters OL, Greiner MA, et al. Initial Thoracic Endovascular Aortic Repair vs Medical Therapy for Acute Uncomplicated Type B Aortic Dissection. JAMA Cardiol. 2023;8(1):44-53.

Round 2

Reviewer 2 Report

 I would like to thank authors for their revised manuscript. They have answered in details all my questions and and they corrected properly the text. No other changes are needed.